# A UHPLC-UV Method Development and Validation for Determining Kavalactones and Flavokavains in *Piper methysticum* (Kava)

**DOI:** 10.3390/molecules24071245

**Published:** 2019-03-30

**Authors:** Yijin Tang, Christine Fields

**Affiliations:** Applied Food Sciences, Inc., 2500 Crosspark Road, Coralville, IA 52241, USA

**Keywords:** *Piper methysticum* (kava), kavalactones, flavokavains, UHPLC-UV, mass spectra, isomerization, single-laboratory validation, quality control

## Abstract

An ultra-high-performance liquid chromatographic (UHPLC) separation was developed for six kava pyrones (methysticin, dihydromethysticin (DHM), kavain, dihydrokavain (DHK), desmethoxyyangonin (DMY), and yangonin), two unidentified components, and three Flavokavains (Flavokavain A, B, and C) in *Piper methysticum* (kava). The six major kavalactones and three flavokavains are completely separated (R_s_ > 1.5) within 15 min using a HSS T3 column and a mobile phase at 60 °C. All the peaks in the LC chromatogram of kava extract or standard solutions were structurally confirmed by LC-UV-MS/MS. The degradations of yangonin and flavokavains were observed among the method development. The degradation products were identified as cis-isomerization by MS/MS spectra. The isomerization was prevented or limited by sample preparation in a non-alcoholic solvent or with no water. The method uses the six kava pyrones and three flavokavains as external standards. The quantitative calibration curves are linear, covering a range of 0.5–75 μg/mL for the six kava pyrones and 0.05–7.5 μg/mL for the three flavokavains. The quantitation limits for methysticin, DHM, kavain, DHK, DMY, and yangonin are approximately 0.454, 0.480, 0.277, 0.686, 0.189, and 0.422 μg/mL. The limit of quantification (LOQs) of the three flavokavains are about 0.270, 0.062, and 0.303 μg/mL for flavokavain C (FKC), flavokavain A (FKA), and flavokavain B (FKB). The average recoveries at three different levels are 99.0–102.3% for kavalactones (KLs) and 98.1–102.9% for flavokavains (FKs). This study demonstrates that the method of analysis offers convenience and adequate sensitivity for determining methysticin, DHM, kavain, DHK, yangonin, DMY, FKA, FKB, and FKC in kava raw materials (root and CO_2_ extract) and finished products (dry-filled capsule and tablet).

## 1. Introduction

Kava (*Piper methysticum*) has been used for a traditional beverage in the Pacific islands, from ancient times, for its relaxant and anxiolytic effects [1,2]. 

The active constituents in kava root have been reported as a group of structurally related lipophilic lactone derivatives, kavalactones (KLs), with an arylethylene-α-pyrone skeleton. More than 18 kavalactones have been isolated from kava [3,4,5], including six major KLs (Figure 1A) as follows: Kavain, 5,5-dihydrokavain (DHK), methysticin, dihydromethysticin (DHM), yangonin, and desmethoxy-yangonin (DMY). Other types of compounds identified in kava include alkaloids, chalcones (flavokavains A, B, and C, Figure 1B), avanones (pinostrobin, 5,7-dimethoxy avanone), cinnamic acid derivatives (bornyl ester of 3,4-methylene dioxycinnamic acid, cinnamic acid bornyl ester), long-chain fatty acids and alcohols, and sterols [6,7,8].

Over a decade ago, potential safety issues with kava applications arose when several cases of liver damage were associated with kava consumption [9,10,11]. Given the concerns around potential liver toxicity with kava usage, the possible mechanisms for hepatotoxicity were investigated [12,13,14]. However, the cause of hepatotoxicity from kava was never clearly associated with one key factor. Kava continues to be consumed on a regular basis within local cultures, with no recent cases of hepatotoxicity reported. Recently, the World Health Organization revisited many of the reported kava-associated hepatotoxicity cases, evaluating the potential causes and providing guidance for safe use as well as recommendations for further studies to identify the potentially hepatotoxic compounds. The chalcone-based flavokavains A (FKA), B (FKB) and C (FKC) were studied with their potential risks and benefits. The studies suggested that the chalcones (FKA, FKB, and FKC), especially FKB, caused hepatotoxicity [15,16,17]. The possible toxicities of flavokavains (FKs) indicated that FKs should be limited to a very low level in the finished products being consumed.

As the main psychoactive components of kava, the contents and chemotypes of six major KLs, are the main criteria of kava beverage quality. Considering the possible hepatotoxicity of FKs, the reliable qualitative and/or quantitative determination of kava compounds is important for the best usage and purification techniques of kava. The RP-HPLC method was considered as an effective method with a good separation and a high accuracy, among many existing analytical methods for determining KLs [18,19,20,21,22]. Most recently, the HPLC method from Meissner and Häberlein [18] and the rapid HPLC method from Brown [22] were proposed for quantifying FKs, along with KLs. The two methods either had a long run time (50 min) or sacrificed the resolution among some key compounds. The UPLC-MS/MS method [23] was also reported for analysis of KLs, however, the linear range of UPLC-MS/MS was limited.

The main goal of this study was to develop and validate a more effective (U)HPLC method for quick and reliable quantification of the six major KLs and three FKs in kava raw roots and rhizomes, as well as finished products, based on the most prevalent instrumentation and standards available. During the method development of this study, it was observed that improper sample preparation of (standard) samples could lead the degradation of yangonin, as well the possible degradations of FKs, utilizing LC-UV-MS. The (U)HPLC methods were fully validated based on the AOAC Guideline [24] for Single-Laboratory Validation of Chemical Methods and ICH Q2 Guideline [25].

## 2. Results and Discussions

### 2.1. Identification of Isomerization of Yangonin and Flavokavains (A, B, and C), along with Sample Preparation

(U)HPLC-UV is still a more applicable method for quantitation of KLs and FKs in most of kava products. In this study, both standards and kava CO_2_ extract samples were applied for the analysis and method development on LC-UV. The chromatographic separation was extended from the previous HPLC study [26] and achieved under a gradient separation at 60 °C. Optimum separation of KLs and FKs was achieved using an UHPLC column (Acquity HSS T3, 100 mm × 2.1 mm, 1.8 μm). Gradient elution was performed using water (100%, no addition of buffer or acids) as solvent A and isopropanol (100%, no addition of buffer or acids) as solvent B, with the gradient program listed in Table 1. The structures of KL or FK compounds were confirmed by the MS/MS spectra (Appendix A).

The two unknown compounds (U1 and U2) reported in a previous study [26] were also presented in the kava CO_2_ extract (Figure 2). The (M + H)^+^ was 261.1 Da for U1 and 263.1 Da for U2, from MS/MS spectra. The MS spectra suggest that the two compounds are very likely to be 5,6-dihydroyangonin (DHY) for U1 and 5,6,7,8-tetrahydroyangonin (THY) for U2, as the literature reported [2]. The structures of THY and DHY are associated with the mass fragmentations of the (M + H)^+^ as 145.1 Da for U1 and 147.1 and 121.1 Da for U2. The two compounds are found as minor KLs in all-natural kava products. The quantitation of the major KLs could be interfered by the two minor components under a low-resolution LC.

The individual stock standard solutions were initially prepared in methanol. Then, the mixed standard solution was diluted with 50% H_2_O/MeOH for LC analysis. Four unknown compounds were observed for the mixed standard solution over several weeks after the preparation. The unknown compounds were with the same molecular weight as both yangonin and flavokavains, but eluted significantly earlier (Figure 3). The fragmentation of the (M + H)^+^ (parent ion) gives an identical product ion spectrum as that of yangonin and flavokavains. The isomerization from yangonin to cis-yangonin in kava standards or extract solutions was reported, especially if aqueous or alcoholic solutions were used [27,28]. MS/MS spectra suggested one of the unknown compounds as cis-yangonin, an isomer of yangonin. MS/MS spectra also indicated the other three compounds related to the isomerization of flavokavains A, B, and C. Those compounds are very likely to be the cis-isomer of flavokavains A, B, and C. For practical reasons, it was assumed that cis products could be formed with an alcoholic solvent or water [27,28]. Many LC methods were attempted for a good resolution (R_s_ > 1.5) for all KLs, especially for the separation of M and DHM. It is very difficult to achieve a good chromatograph separation (R_s_ > 1.5) for all KLs and FKs with the additions of cis-yangonin and cis-FKs. It is suggested that during validation of the proposed method, the stock standard solutions of KLs and FKs are prepared in a non-alcoholic solvent, like acetonitrile. The kava working standard solution and sample solution should be freshly prepared for LC analysis, to prevent or minimize the isomerization of yangonin and FKs.

### 2.2. Method Validation

#### 2.2.1. Specificity/Resolution

Each individual reference standard was injected into the HPLC-UV to compare with the standard mixture (Appendix A). Identification of KLs in the test materials was determined by comparing peak retention times and UV spectra to the reference standards. Representative chromatograms of the standard mixture and kava products are displayed in Figure 4. Under the chromatographic conditions used in the present study, all six major KLs and three FKs were eluted separately following this following order: Methysticin, DHM, kavain, DHK, yangonin, DMY, FKC, FKA, and FKB. Values for the relative retention times (α), retention factors (*k′*), and chromatographic resolutions (R_s_), calculated from the LC analysis of the kava standard mixtures (Figure 4), are summarized in Table 2. Retention factors (*k′*) were calculated as *k′* = (*t_r_* − *t*_0_)/*t*_0_, where *t_r_* is the retention time of the analyte and *t*_0_ is the retention time of unretained compounds (solvent front). The *k′* values were within the optimum range (*k′* > 2) for satisfactory chromatographic elution. An excellent chromatographic specificity was observed with the good resolution of the peaks (R_s_ > 1.5) and with no significant interfering peaks for all compounds in the mixed standard sample. The total chromatography run time was 15.5 min.

#### 2.2.2. Standard Linearity

To evaluate the linearity of the calibration curves, calibration standard mixed solutions, at the target concentrations, were prepared as described above. Each peak area of the chromatograms was recorded as the UV responses at 239 nm for methysticin, DHM, kavain, and DHK, and at 350nm for yangonin, DMY, and FKs. Calibration curves were plotted by the peak area vs. the concentration of the standard compounds (Appendix A). Regression analyses were processed by Lab Solution software. Calibration curves were linear over the concentration range used, with an R^2^ > 0.999. The normalized intercept/slope of the regression line and the correlation coefficient were calculated for the whole data set. The method was evaluated by determining the coefficient of linearity and the intercept values, as summarized in Table 3.

#### 2.2.3. Limit of Detection and Limit of Quantification

The limit of detection (LOD) and the limit of quantification (LOQ) of the kava standard assay were determined by the standard deviation of the y-intercepts over the slope of the regression lines from three replicated calibration curves on different days. The LODs and LOQs for six major KLs and three FKs were calculated and summarized in Table 3.

#### 2.2.4. Recovery 

A spike recovery study based on spiking six kavalactones and three flavokavains into a kava test material (kava root) at high, medium, and low levels, as well as non-spiked, were completed and shown in Table 4. The amounts of kavalactones and flavokavains were compared before and after spiking. As the AOAC guidelines suggest for single-laboratory validation, the recoveries were calculated in two ways, as follows: (1) Total recovery based on recovery of the native plus added analyte, and (2) marginal recovery based only on the added analyte (the native analyte is subtracted from both the numerator and denominator). The total recovery is used for the native analytes presented in amounts greater than about 10% of the amount added, otherwise, the marginal recovery is applied. The average recoveries for each analyte at each level were 99.0–102.3% for KLs and 98.1–102.9% for FKs, within the ranges of recovery limits from 95% to 102% at 10% concentration, 92% to 105% at 1% concentration, 90% to 108% at 0.1% concentration, and 85% to 110% for 0.01 at 0.01% concentration.

#### 2.2.5. Precision 

The precision and accuracy of the method was assessed by determining the intraday precisions (n = 5) from repeating the analysis of the kava samples on the same day; and the interday precisions (n = 3 × 5, overall 15) from analyzing the same kava samples over the different days. Both intraday and interday precisions were calculated as RSD_r_ (%) = (standard deviation)/(mean) × 100. As shown in Table 5, all nine analytes had adequate precision in each of the four different solid matrices at different concentrations. The level of KLs were about 10–100 mg/g for CO_2_ extract and root. The level of FKs in the kava CO_2_ extract and the root were about 0.13–0.30 mg/g for FKC, 0.60–1.5 mg/g for FKA, and 0.50–0.90 mg/g for FKB. The intraday repeatability relative standard deviations (RSD_r_) of the CO_2_ extract and the root were 0.31% to 1.61% for KLs and 0.98% to 3.83% for FKs, which are <2% for KLs and <4% for FKs, as AOAC guidelines suggested. About 2–10 mg/g for KLs, 0.08–0.10 mg/g for FKC, 0.46–0.75 mg/g for FKA, and 0.60–1.00 mg/g for FKB were detected for kava products in tablets and capsules. The intraday RSD_r_ of kava products in tablets and capsules were 0.28% to 1.96% for KLs and 0.23% to 4.24% for FKs, which are also <3% for KLs and <6% for FKs, as AOAC guidelines suggested. Overall, the interday repeatability relative standard deviations (RSD_r_) ranged from 0.50% to 2.56% for kavalactones and 2.44% to 5.52% for flavokavains. The Horwitz ratio (HorRat) values are used to evaluate method performance based on the ratios of actual precision to predicted precision. AOAC guidelines for single-laboratory validation accept a HorRat range from 0.5 to 2. In our method, the HorRat value for kavalactones ranged from 0.24 to 1.05 and for flavokavains ranged from 0.77 to 1.89. A HorRat value lower than 0.5 was considered acceptable, considering the analysis was performed under tightly controlled conditions.

## 3. Discussion

As a more applicable method, the RP-HPLC method was reported for quantitation analysis of kava products [18,19,20,21,22,26]. Over many studies, Alexander H. Schmidt and Imre Molnar [26] applied computer-assisted optimization in the development to achieve a great resolution for the separations of all the major KLs. The study also included the two minor KLs (DHY and THY), but not FKs. As the above description, the kava root (~ 10% KLs), the kava CO_2_ extract product (~ 20% KLs), and commercial tablet and capsule kava products were analyzed for the quantitation of each kava compound, based on the standard responses of M, DHM, K, and DHK (at 239 nm) and Y, DMY, and FKs (at 350 nm). The calculated concentrations are shown in Table 5. In this study, the percentage of each KL and FK over the total KLs and FKs, calculated as follows: (1)$$\frac{x}{M+\mathrm{DHM}+K+\mathrm{DHK}+Y+\mathrm{DMY}+\mathrm{FKs}}\times 100$$

The kavalactone and flavokavain profile were normalized to the percentage of total kavalactone content and presented in Figure 5. Although there is a large difference in the total content of kava lactones, the relative intensities of the individual compounds differ only slightly. AFS kava root and the CO_2_ extract contain a very similar profile of KLs at different levels, about 100 mg KLs per gram in the root and at about 200 mg KLs per gram in the CO_2_ extract. The capsule and tablet products contain about 40 and 30 mg KLs per gram.

The quality of kava products was considered as a safety issue for consumers. The key criteria for the quality of kava product are the contents and chemotypes of six major KLs. Different quantity and ratio of KLs can impact their physiological action and safety [29]. The chemotypes of kava products were identified as noble or non-noble varieties following the simple system described by Lebot and Lévesque [30,31]. The six major KLs are used to define the chemotype (1 = DMY; 2 = DHK; 3 = yangonin; 4 = kavain; 5 = DHM; and 6 = methysticin). The different chemotypes of kava products are coded by listing, in decreasing order of proportion, the KLs. The noble cultivars have chemotypes rich in kavain, like 423561 or 423651. The chemotypes of 521634, 526341, or 254631 represent non-noble cultivars with very high proportions of DHM or DHK.

In this study, the chemotypes of kava products were determined by the percentage of each KL. The results show that the chemotypes of kava root (462351) were very similar to the CO_2_ extract (463251), while the chemotypes of the capsule (245631) were very similar to the tablet (246531). 

Due to the possible hepatotoxicities of FKs, another key criterion for the quality of kava products is to limit the amount of FKs, as FKs/KLs ≤ 0.29 and FKB% < 0.15% [16,17]. The ratios of FKs/KLs for kava products in this study were detected as 0.013 for the CO_2_ extract, 0.014 for the root, 0.041 for the capsule, and 0.044 for the tablet. FKB was 0.086%, 0.055%, 0.06% and 0.096% for the kava CO_2_ extract, the root, the tablets, and the capsules, respectively. In this study, it was noticed that cis-isomers of FKA and FKB were present at different levels in the powdered samples (Figure 4). The level of cis-isomers of FKA and FKB were found to be very low in both the kava CO_2_ extract and the kava root, while a small amount of cis-isomers of FKA and FKB were observed in the capsule and tablet kava products. 

## 4. Materials and Methods 

### 4.1. Chemicals and Materials 

The 2-propanol (HPLC grade), acetonitrile (ACN, HPLC grade), water (H2O, LC-MS grade), and methanol (MeOH, LC-MS grade) were purchased from Fisher Scientific (Hampton, NH, USA). The reference standard compounds of D, l-kavain (purity: 95% HPLC), yangonin (purity: 95% HPLC), flavokavain A (purity: 95% HPLC), flavokavain B (purity: 95% HPLC), and flavokavain C (purity: 95% HPLC) were purchased from Extrasynthese (Genay, France). The standard compounds of methysticin (purity: 99.63% HPLC), dihydromethysticin (purity 98.68% HPLC), desmethoxyyangonin (purity: 97.84% HPLC), and dihydrokavain (purity: 99.04% HPLC) products were from PhytoLab (Vestenbergsgreuth, Germany) and purchased from Cerilliant (Round Rock, TX, USA). Ultrapure (18 MΩ) water was produced using a Barnstead™ GenPure™ Pro Water Purification System from Thermo Fisher Scientific (Waltham, MA, USA). 

### 4.2. Instrumentation

Method development and validation studies were performed on a Shimadzu Nexera-X2 UHPLC system (Shimadzu Scientific Instruments, Columbia, MD, USA), equipped with a LC-30AD pump, a SIL-30AC autosampler with a thermostated unit, a thermostated column compartment, and an SPD-M30A PDA detector. The UHPLC system was also interfaced with tandem Q-Exactive Orbitrap mass spectrometer (Thermo Fisher Scientific Inc., San Jose, CA, USA). High-resolution MS and MS^2^ spectra were obtained on the Q-Exactive Orbitrap mass spectrometer equipped with a heated electrospray ionization, operated in both positive and negative ion mode. The optimized parameters were set as follows: Capillary voltage, 3.0 kV; sheath gas flow rate, 35 arbitrary unit; auxiliary gas flow rate, 5 arbitrary unit; sweep gas flow rate, 5 arbitrary unit; capillary temperature, 325 °C; and sheath gas heater temperature, 200 °C. MS scans were recorded in a mass range of *m*/*z* 100–1500 at a resolution of 70,000 with an AGC target of 3 × 10^6^. After each MS scan, up to 5 of the most abundant multiply charged ions were selected for fragmentation. MS^2^ scans were recorded in a mass range of *m*/*z* 50 to the parent ion at a resolution of 17,500, with an AGC target of 1 × 10^5^ and a maximum fill time of 50 ms, using the stepped NCE of 25 and 35 for fragmentation in the HCD cell. Data were acquired from 50 to 1500 Da with dd-MS^2^ or MS^2^ in centroid mode. Raw data were acquired and processed using the Xcalibur software (Version 2.3.1, Thermo Electron Corporation, San Jose, CA, United States). 

### 4.3. Chromatographic Condition

The chromatographic separation was extended from the previous HPLC study [24] and achieved under a gradient separation at 60 °C. Optimum separation of KLs and FKs was achieved using an UHPLC column (Acquity HSS T3, 100 mm × 2.1 mm, 1.8 μm). Gradient elution was performed using water (solvent A) and isopropanol (solvent B) with the gradient program listed in Table 1.

### 4.4. Standard Preparation

Individual standard stock solutions of 1000 µg/mL of each kava standard compound were accurately prepared by weighing about 10 mg of each compound and dissolving them into a 10 mL volumetric flask using acetonitrile. Volumetric flasks were sonicated for 10 min and wrapped with aluminum foil to protect them from light. Stock solutions were kept refrigerated. Working standard solutions were prepared fresh on a daily basis by pipetting aliquots of stock solutions and serial dilutions with 50% ACN were made at concentrations ranging from 0.5 µg/mL to 75 µg/mL for KLs and from 0.05 µg/mL to 7.5 µg/mL for FKs.

### 4.5. Test Materials and Sample Preparation

The kava CO_2_ extract and root were obtained from Applied Food Sciences Inc. (AFS, Austin, TX, USA). Kava commercial products (dry-filled capsules and formulated tablets) were purchased from Foods market. All the samples were analyzed more than triplicate, unless stated otherwise.

#### 4.5.1. Kava CO_2_ Extract and Root Powders

A total of 100 mg of kava CO_2_ extract or 300 mg of the root powder were first extracted with 15 mL ACN and sonicated for 30 min at 40 °C in a Fisher sonication bath. Following a 10-min centrifugation at 12,000× *g*, the supernatant was transferred to a 50 mL volumetric flask. The remaining residue was re-extracted twice with 15 mL ACN, following the same procedure. The volumetric flask was filled to the mark with ACN in the end. The samples were freshly diluted 1× with 18MW water and the extracts were filtered through a 3 mm syringe fitted with a 0.22 µm nylon filter (VWR) into an amber glass HPLC vial and readied for LC analysis.

#### 4.5.2. Capsules and Phytocaps

The content of the 20 capsules or phytocaps were combined and mixed thoroughly. Two hundred milligrams of the capsules or phytocaps content were extracted with ACN and acetone following the same procedure as above.

### 4.6. Method Validation Parameters

This method was validated following the AOAC and ICH (Q2) guidelines [24,25] for conducing single-laboratory validation. For all the standards, 1000 µg/mL stock solutions were prepared by dissolving individual reference materials in ACN in volumetric flasks. The stock solutions of the reference materials were stored at −20 °C for long-term storage. The stock solution for each standard was mixed to the kava working standard solution at the concentration 100 µg/mL for KLs and 10 µg/mL for FKs, then diluted to the appropriate concentration to establish the retention time and combined at different concentration levels for external calibration.

#### 4.6.1. Specificity/Resolution

The mixed reference standard was injected into the HPLC-UV to establish the selectivity of the method. The resolution for each reference standard was calculated. The value of R_s_ > 1.5 between closely eluting components was considered acceptable for FKs and major KLs.

#### 4.6.2. Linearity

The linearity for the reference standard was determined by seven-point standard calibration curves. The standard curve for the six KLs ranged from 0.5 µg/mL to 75 µg/mL (0.5, 1.0, 5.0, 10.0, 25.0, 50.0, and 75.0 µg/mL). The standard curve for the three FKs ranged from 0.05 µg/mL to 7.5 µg/mL (0.05, 0.1, 0.5, 1.0, 2.5, 5.0, and 7.5 µg/mL). A simple linear regression was used to calculate R^2^ value, the slope, and the y-intercept of each curve for each analyte. An R^2^ ≥ 99.9% value was considered acceptable. The calibration standards of seven KLs and three FKs were triplicated at the seven concentrations and analyzed over three days.

#### 4.6.3. LOD and LOQ

The limit of detection (LOD) and limit of quantification (LOQ) of the kava standard assay were determined from the calibration curve method, as ICH Q2 (R1) recommendations [25], analyzing at least three replicates of the calibration standards. The LOD and LOQ of the proposed method were calculated using the following equations:(2)$$\mathrm{LOD}=\frac{3.3\times \mathrm{Stdev}\;y–\mathrm{intercept}\;\mathrm{of}\;\mathrm{Calibration}\;\mathrm{Curve}}{\mathrm{Slope}\;\mathrm{of}\;\mathrm{calibration}\;\mathrm{curve}\;\left(A_{\mathrm{ave}}\right)}$$
(3)$$\mathrm{LOQ}=\frac{10\times \mathrm{Stdev}\;y–\mathrm{intercept}\;\mathrm{of}\;\mathrm{Calibration}\;\mathrm{Curve}}{\mathrm{Slope}\;\mathrm{of}\;\mathrm{calibration}\;\mathrm{curve}\;\left(A_{\mathrm{ave}}\right)}$$

#### 4.6.4. Recovery

Spike recovery experiments were performed at three levels (high, 30 µg/mg; medium, 15 µg/mg; and low, 2.5 µg/mg) for KLs and three levels (high, 3 µg/mg; medium, 1.5 µg/mg; and low, 0.25 µg/mg) for FKs. Powdered kava root material was analyzed for KLs and FKs prior to the standards being spiked. The appropriate amount of reference standards was used to spike the powdered kava root material, followed by the extraction process. Considering the cost of the reference standards, for high-level spike recovery experiments a 5 mg sample was extracted with 5 mL extraction solvent (ACN). For medium-and low-level spike recovery experiments, a 10 mg sample was extracted with 10 mL extraction solvent (ACN). Three replicates were performed at each level and the mean recovery was calculated.

#### 4.6.5. Precision

Four independent replicates of the same sample were prepared and analyzed on three separate days (n = 5 × 3). The within-day, between-day, overall precision for all nine target compounds were calculated for single-laboratory validation.

## 5. Conclusions

In conclusion, the UHPLC-UV method described herein for the determination of six major KLs and three FKs in kava raw materials and finished products was validated based on AOAC Guidelines for Single-Laboratory Validation of Chemical Methods for Dietary Supplements and Botanicals. This method maximized efficiency and chromatographic resolution under short analysis times. Both DHY and THY were found in all the kava products. The two minor KLs were well-separated from the major KLs under the excellent LC resolution. The isomerizations of yangonin and FKs were prevented or limited by the usage of non-alcoholic solvents, like acetonitrile, for sample preparation in this method. This suggested method of kava analysis is free of interference from minor KLs, like THY & DHY, and at a very low level of interferences from cis-isomers of yangonin and FKs. The results of the study demonstrate that this UHPLC-UV analytical method is a successful approach to determine methysticin, DHM, kavain, DHK, yangonin, DMY, FKA, FKB, and FKC in kava raw materials (kava root powder and kava CO_2_ extract) and finished products (dry-filled capsules or formulated tablets) under a quick analysis time of 15 min and it therefore expands the scope to analyze a broad variety of market samples. 

## Figures and Tables

**Figure 1 molecules-24-01245-f001:**
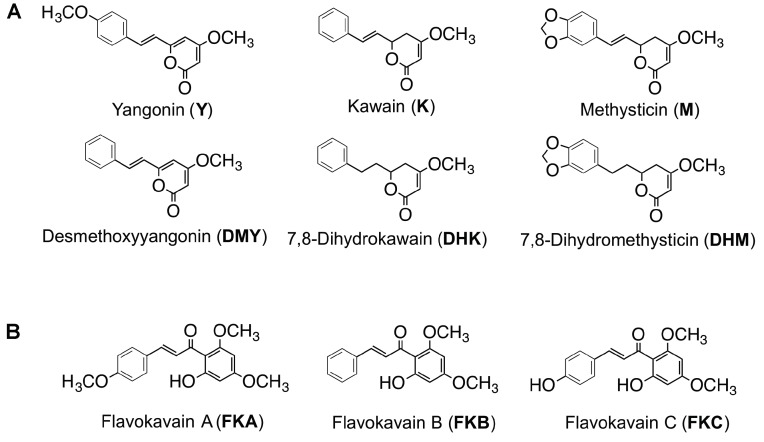
Chemical structures of (**A**) six major kavalactones and (**B**) three flavokavains.

**Figure 2 molecules-24-01245-f002:**
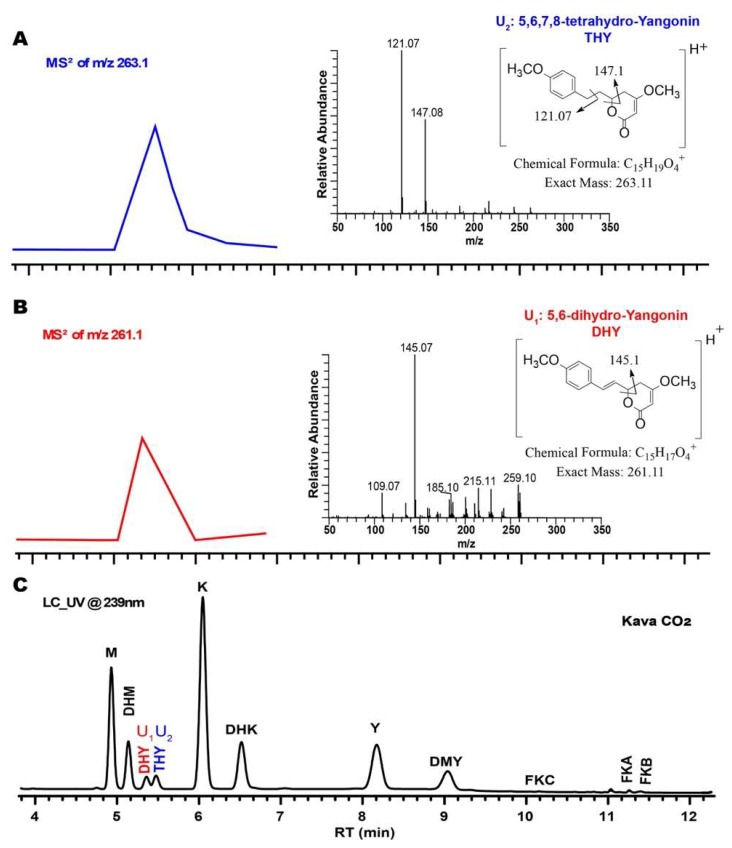
UHPLC-UV-MS data of kava CO_2_ extract as follows: (**A**) MS/MS^2^ trace of *m*/*z* 263.1 and MS^2^ spectrum of peak U_2_ (5,6,7,8-Tetrahydroyangonin, THY); (**B**) MS/MS^2^ trace of *m*/*z* 261.1 and MS^2^ spectrum of peak U_1_ (5,6-Dihydroyangonin, DHY); (**C**) UV trace at the wavelength of 239 nm; M, methysticin; DHM, 7,8-Dihydromethysticin; K, kavain; DHK, 7,8-Dihydrokavain; Y, yangonin; DMY, desmethoxyyangonin; FKA (B or C) flavokavains A (B or C).

**Figure 3 molecules-24-01245-f003:**
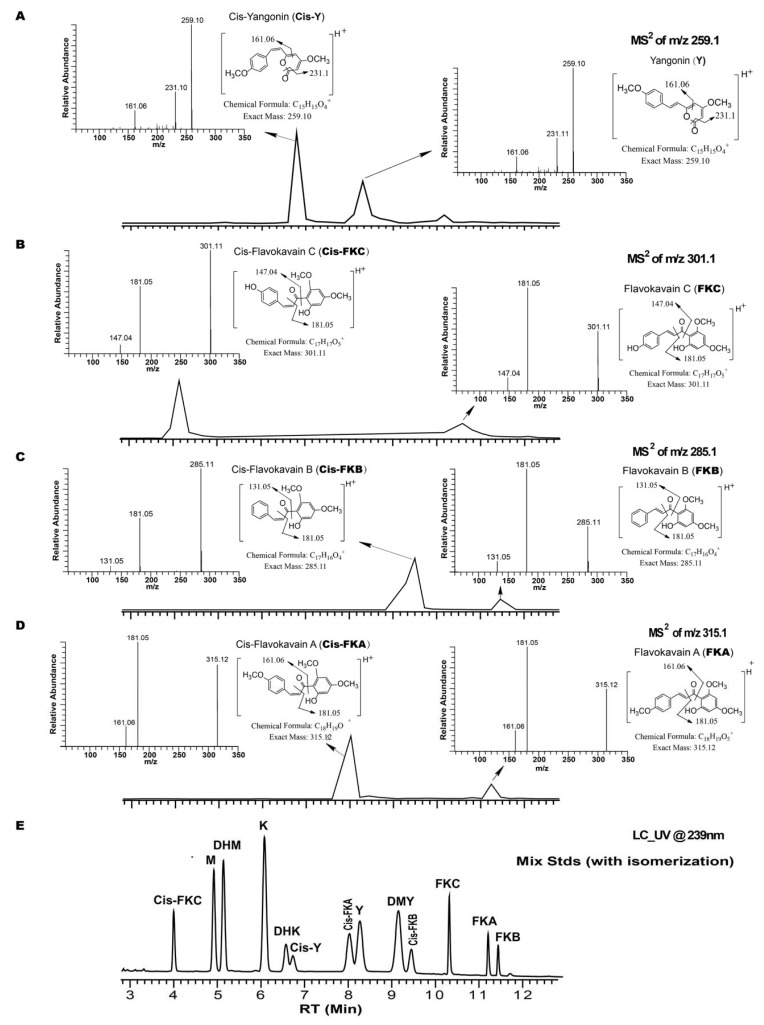
UHPLC-UV-MS data of degraded kava standard mixture, as follows: (**A**) MS/MS^2^ trace of *m*/*z* 259.1, and MS^2^ spectra of peaks of Y and cis-Y; (**B**) MS/MS^2^ trace of *m*/*z* 301.1, and MS^2^ spectra of peaks of FKC and cis-FKC; (**C**) MS/MS^2^ trace of *m*/*z* 285.1, and MS^2^ spectra of peaks of FKB and cis-FKB; (**D**) MS/MS^2^ trace of *m*/*z* 315.1 and MS^2^ spectra of peaks of FKA and cis-FKA; (**E**) UV trace at the wavelength of 239 nm; M, methysticin; DHM, 7,8-Dihydromethysticin; K, kavain; DHK, 7,8-Dihydrokavain; Y, Yangonin; Cis-Y, cis-yangonin; DMY, Desmethoxyyangonin; FKA (B or C), flavokavains A (B or C); cis-FKA (B, or C), flavokavains A (B or C).

**Figure 4 molecules-24-01245-f004:**
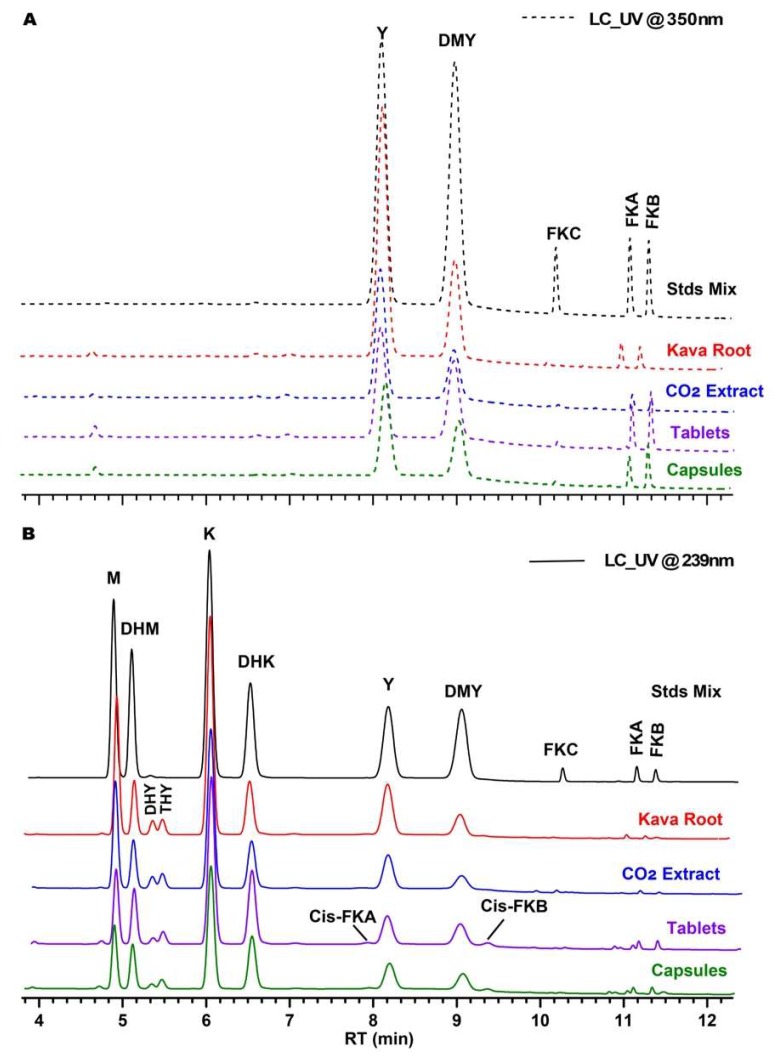
UHPLC chromatography of kava standard mixture and kava products. (**A**) UV trace at the wavelength of 350 nm; (**B**) UV trace at the wavelength of 239 nm; M, methysticin; DHM, 7,8-Dihydromethysticin; DHY, 5,6-Dihydroyangonin; THY, 5,6,7,8-Tetrahydroyangonin; K, kavain; DHK, 7,8-Dihydrokavain; DMY, Desmethoxyyangonin; Y, Yangonin; FKA (B or C), flavokavains A (B or C); cis-FKA (B), cis-flavokavains A (B).

**Figure 5 molecules-24-01245-f005:**
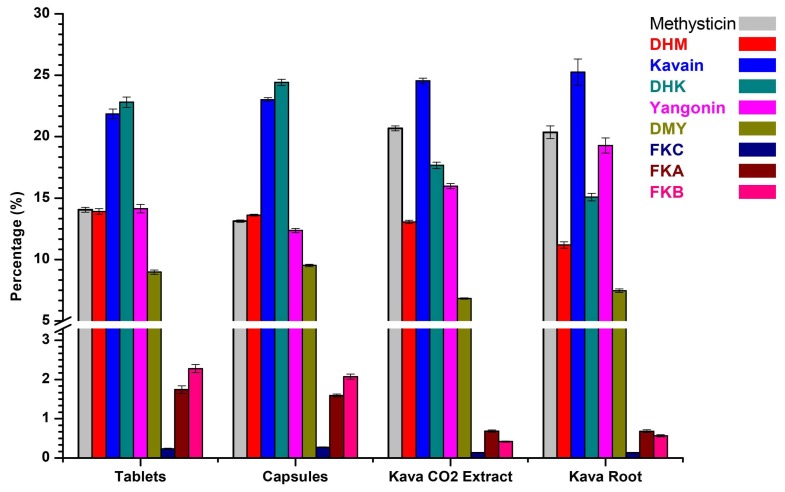
The profiles of kavalactones and flavokavains for kava products.

**Table 1 molecules-24-01245-t001:** Gradient UHPLC elution profile.

Time/min	A (H_2_O, %)	B (IPA, %)	Flow Rate (mL/min)
0.00	95	5	0.50
0.35	90	10	0.50
0.39	78	22	
2.00	78	22	0.43
7.00	78	22	0.41
7.50	71	29	0.41
10.50	25	75	0.41
11.50	0	100	0.41
12.80	0	100	0.41
13.00	95	5	0.41
15.00	95	5	0.44
15.50	End		0.50

**Table 2 molecules-24-01245-t002:** Parameters of kavalactones and flavokavains.

Compound	Retention Factor (*k*′)	Relative RT (*α*)	Chromatographic Resolution (*R_s_*)
**Methysticin (1)**	8.01	–	–
**Dihydromethysticin (2)**	8.41	α_1/2_ = 1.045	R_1/2_ = 1.78
**Kavain (3)**	10.15	α_2/3_ = 1.184	R_2/3_ = 6.61
**Dihdyrokavain (4)**	11.07	α_3/4_ = 1.082	R_3/4_ = 3.02
**Yangonin (5)**	14.12	α_4/5_ = 1.253	R_4/5_ = 8.20
**Desmethoxyyangonin (6)**	15.76	α_5/6_ = 1.109	R_5/6_ = 3.62
**Flavokavain C (7)**	18.18	α_6/7_ = 1.144	R_6/7_ = 7.67
**Flavokavain A (8)**	19.86	α_7/8_ = 1.088	R_7/8_ = 11.92
**Flavokavain B (9)**	20.29	α_8/9_ = 1.020	R_8/9_ = 3.04

**Table 3 molecules-24-01245-t003:** Calibration parameters for kavalactones and flavokavains from three different calibration curves.

	Calibration Range (μg/mL)	Slope (±SD) ^a^	Y-Intercept (±SD) ^a^	r^2^ (±SD) ^a^	LOD (μg/mL)	LOQ (μg/mL)
Methysitcin	0.50~75.0	106708.0 ± 700.0	−3443.1 ± 4846.8	0.99996 ± 0.00005	0.150	0.454
DHM		80132.1 ± 502.3	6666.6 ± 3847.5	0.99997 ± 0.00002	0.158	0.480
Kavain		176782.0 ± 1450.0	−5987.0 ± 4889.8	0.99996 ± 0.00004	0.091	0.277
DHK		80884.5 ± 381.5	11433.5 ± 5551.5	0.99995 ± 0.00002	0.226	0.686
Yangonin		167271.0 ± 1964.9	3729.7 ± 3167.9	0.99994 ± 0.00003	0.062	0.189
DMY		164787.0 ± 2352.4	−12618.3 ± 6952.1	0.99995 ± 0.00003	0.139	0.422
FKC	0.05~7.50	135720.0 ± 3576.4	−3194.0 ± 3663.5	0.99970 ± 0.00017	0.089	0.270
FKA		143817.0 ± 2873.9	768.8 ± 889	0.99980 ± 0.00018	0.020	0.062
FKB		150461.0 ± 3339.0	−1374.6 ± 4555.5	0.99988 ± 0.00017	0.100	0.303

Note: Calibration curves were performed three times on different days and established by measuring the concentration vs. the corresponding peak area. The given values are the mean of three replicates ± standard deviation ^a^.

**Table 4 molecules-24-01245-t004:** Spike recovery results of HPLC-UV method for determination of kavalactones and flavokavains.

	Methysticin	DHM	Kavain	DHK	DMY	Yangonin	FKC	FKA	FKB
Spiked Level	3.0%						~0.3%		
Native (μg/mL) ^a^	40.16 ± 0.62	22.14 ± 0.34	47.80 ± 0.74	29.75 ± 0.46	36.94 ± 0.57	14.52 ± 0.22	0.296 ± 0.005	1.27 ± 0.02	1.03 ± 0.02
Spiked (μg/mL) ^b^	60.02 ± 0.80	58.73 ± 0.78	59.28 ± 0.79	59.23 ± 0.79	59.36 ± 0.79	57.73 ± 0.77	5.67 ± 0.08	7.99 ± 0.11	5.76 ± 0.08
After Spiked (μg/mL) ^b^	100.2 ± 0.2	80.87 ± 0.44	107.1 ± 0.1	88.99 ± 0.33	96.30 ± 0.22	72.24 ± 0.54	5.97 ± 0.07	9.25 ± 0.09	6.79 ± 0.06
Detected (μg/mL) ^a^	101.5 ± 0.3	81.43 ± 0.51	109.6 ± 0.6	89.63 ± 0.44	98.21 ± 0.41	73.42 ± 0.36	5.86 ± 0.16	9.48 ± 0.14	6.92 ± 0.11
Marginal recovery (%) ^a^	102.1 ± 0.1	101.0 ± 0.1	104.2 ± 1.0	101.1 ± 0.4	103.2 ± 0.6	102.0 ± 0.4	**98.1 ± 1.6** ^c^	**102.9 ± 0.6**	**102.2 ± 0.8**
Total Recovery (%) ^a^	**101.3 ± 0.1**	**100.7 ± 0.1**	**102.3 ± 0.5**	**100.7 ± 0.3**	**102.0 ± 0.4**	**101.6 ± 0.3**	98.2 ± 1.5	102.5 ± 0.5	101.3 ± 0.1
Spiked Level	1.5%						~0.15%		
Native (μg/mL) ^a^	41.04 ± 0.81	22.63 ± 0.45	48.86 ± 0.97	30.41 ± 0.60	37.76 ± 0.75	14.84 ± 0.29	0.302 ± 0.006	1.29 ± 0.03	1.05 ± 0.02
Spiked (μg/mL) ^b^	30.21 ± 0.15	29.55 ± 0.15	29.83 ± 0.15	29.81 ± 0.15	29.87 ± 0.15	29.05 ± 0.14	2.86 ± 0.014	4.02 ± 0.02	2.90 ± 0.01
After Spiked (μg/mL) ^b^	71.25 ± 0.66	52.18 ± 0.30	78.69 ± 0.82	60.22 ± 0.46	67.63 ± 0.60	43.89 ± 0.15	3.15 ± 0.01	5.31 ± 0.01	3.95 ± 0.01
Detected (μg/mL) ^a^	70.50 ± 0.74	51.73 ± 0.42	78.02 ± 1.17	59.63 ± 0.51	67.21 ± 0.83	44.12 ± 0.36	3.10 ± 0.02	5.36 ± 0.02	3.97 ± 0.02
Marginal recovery (%) ^a^	97.4 ± 2.5	99.0 ± 1.6	93.8 ± 2.5	98.6 ± 1.1	97.8 ± 2.9	100.8 ± 0.9	98.4 ± 0.5	101.0 ± 0.5	100.3 ± 0.9
Total Recovery (%) ^a^	**99.0 ± 1.1**	**99.1 ± 0.9**	**99.2 ± 1.0**	**99.0 ± 0.5**	**99.4 ± 1.3**	**100.5 ± 0.6**	**98.1 ± 0.5**	**100.8 ± 0.4**	**100.3 ± 0.7**
Spiked Level	0.25%						~0.025%		
Native (μg/mL) ^a^	40.83 ± 0.91	22.51 ± 0.50	48.60 ± 1.09	30.25 ± 0.68	37.56 ± 0.84	14.76 ± 0.33	0.300 ± 0.007	1.29 ± 0.03	1.05 ± 0.02
Spiked (μg/mL) ^b^	5.01 ± 0.07	4.90 ± 0.07	4.95 ± 0.07	4.94 ± 0.07	4.95 ± 0.07	4.82 ± 0.07	0.473 ± 0.007	0.67 ± 0.01	0.48 ± 0.01
After Spiked (μg/mL) ^b^	45.84 ± 0.85	27.41 ± 0.44	53.55 ± 1.02	35.20 ± 0.62	42.51 ± 0.78	19.58 ± 0.27	0.774 ± 0.004	1.95 ± 0.02	1.53 ± 0.02
Detected (μg/mL) ^a^	45.51 ± 1.06	27.45 ± 0.51	53.47 ± 1.30	35.25 ± 0.78	42.21 ± 1.06	19.66 ± 0.32	0.779 ± 0.002	1.96 ± 0.03	1.53 ± 0.03
Marginal recovery (%) ^a^	93.6 ± 4.8	100.7 ± 1.4	98.5 ± 5.7	101.1 ± 3.6	93.9 ± 5.6	101.7 ± 1.1	101.0 ± 0.7	101.7 ± 1.5	99.9 ± 3.2
Total Recovery (%) ^a^	**99.3 ± 0.5**	**100.1 ± 0.2**	**99.9 ± 0.5**	**100.1 ± 0.5**	**99.3 ± 0.7**	**100.4 ± 0.3**	**100.6 ± 0.4**	**100.6 ± 0.5**	**100.0 ± 1.0**

**Note:** The given values are the mean of three replicated measurements ± standard deviation ^a^. The values for all the spiked and after-spiked concentrations were calculated as the mean of the three replicated plus the standard deviation ^b^. The bold recoveries were applied for the final evaluation ^c^.

**Table 5 molecules-24-01245-t005:** Precision summary of the HPLC-UV method for detecting kavalactones and flavokavains in kava products.

		Methysticin	DHM	Kavain	DHK	Yangonin	DMY	FKC	FKA	FKB
		Tablets								
LOQ (mg/g)		0.076	0.080	0.046	0.114	0.032	0.070	0.045	0.010	0.051
D1	Mean ± SD (mg/g)	3.79 ± 0.03	3.94 ± 0.03	6.71 ± 0.07	7.17 ± 0.06	3.55 ± 0.03	2.76 ± 0.03	0.082 ± 0.001	0.477 ± 0.007	0.627 ± 0.008
	RSD_r_ (%) Intraday	0.86	0.80	1.06	0.82	0.98	1.08	1.28	1.39	1.32
D2	Mean ± SD (mg/g)	3.83 ± 0.01	3.96 ± 0.01	6.67 ± 0.03	7.05 ± 0.04	3.58 ± 0.04	2.78 ± 0.04	0.076 ± 0.001	0.452 ± 0.002	0.587 ± 0.002
	RSD_r_ (%) Intraday	0.34	0.34	0.45	0.50	1.22	1.30	1.64	0.49	0.28
D3	Mean ± SD (mg/g)	3.83 ± 0.02	3.96 ± 0.01	6.68 ± 0.02	7.06 ± 0.04	3.64 ± 0.03	2.75 ± 0.02	0.078 ± 0.001	0.459 ± 0.001	0.592 ± 0.002
	RSD_r_ (%) Intraday	0.39	0.28	0.32	0.60	0.81	0.70	0.77	0.23	0.49
Mean ± SD (mg/g)		3.81 ± 0.03	3.95 ± 0.02	6.69 ± 0.05	7.09 ± 0.07	3.59 ± 0.05	2.77 ± 0.03	0.079 ± 0.003	0.463 ± 0.011	0.602 ± 0.019
RSD (%) Interday		0.73	0.54	0.68	1.04	1.40	1.05	3.75	2.44	3.20
HorRat		0.32	0.24	0.32	0.50	0.60	0.43	0.91	0.77	1.05
		Capsules								
LOQ (mg/g)		0.151	0.160	0.092	0.229	0.063	0.141	0.090	0.021	0.101
D1	Mean± SD (mg/g)	5.87 ± 0.07	5.80 ± 0.05	9.09 ± 0.09	9.48 ± 0.11	5.85 ± 0.09	3.74 ± 0.05	0.103 ± 0.002	0.702 ± 0.008	0.920 ± 0.010
	RSD_r_ (%) Intraday	1.14	0.82	0.97	1.20	1.50	1.33	2.18	1.16	1.08
D2	Mean± SD (mg/g)	5.98 ± 0.07	5.93 ± 0.08	9.33 ± 0.12	9.73 ± 0.14	6.06 ± 0.10	3.83 ± 0.07	0.094 ± 0.004	0.773 ± 0.011	0.999 ± 0.019
	RSD_r_ (%) Intraday	1.10	1.31	1.29	1.45	1.66	1.74	4.24	1.39	1.93
D3	Mean± SD (mg/g)	5.99 ± 0.12	5.98 ± 0.11	9.37 ± 0.12	9.85 ± 0.13	6.07 ± 0.10	3.87 ± 0.06	0.101 ± 0.002	0.796 ± 0.011	1.012 ± 0.012
	RSD_r_ (%) Intraday	1.96	1.82	1.32	1.32	1.59	1.58	2.04	1.43	1.21
Mean± SD (mg/g)		5.95 ± 0.10	5.90 ± 0.11	9.26 ± 0.17	9.69 ± 0.20	6.00 ± 0.14	3.81 ± 0.08	0.099 ± 0.004	0.757 ± 0.042	0.977 ± 0.044
RSD_r_ (%) Interday		1.67	1.84	1.80	2.06	2.26	2.06	4.51	5.56	4.49
HorRat		0.77	0.85	0.89	1.03	1.05	0.89	1.13	1.89	1.59
		CO_2_ Extract								
LOQ (mg/g)		0.454	0.480	0.277	0.686	0.189	0.422	0.270	0.062	0.303
D1	Mean± SD (mg/g)	42.39 ± 0.41	26.75 ± 0.29	50.5 ± 0.7	36.2 ± 0.5	32.5 ± 0.3	14.0 ± 0.1	0.29 ± 0.01	1.34 ± 0.02	0.84 ± 0.02
	RSD_r_ (%) Intraday	0.97	1.08	1.28	1.39	1.06	0.99	2.13	1.84	2.11
D2	Mean± SD (mg/g)	42.81 ± 0.21	27.0 ± 0.2	50.6 ± 0.3	36.2 ± 0.4	33.1 ± 0.2	14.2 ± 0.1	0.29 ± 0.01	1.43 ± 0.02	0.85 ± 0.01
	RSD_r_ (%) Intraday	0.50	0.54	0.55	0.97	0.56	0.38	2.09	1.34	1.22
D3	Mean± SD (mg/g)	42.75 ± 0.53	27.1 ± 0.3	50.8 ± 0.5	36.9 ± 0.4	33.3 ± 0.2	14.1 ± 0.1	0.270 ± 0.004	1.47 ± 0.02	0.89 ± 0.03
	RSD_r_ (%) Intraday	1.25	1.08	0.88	1.18	0.74	0.68	1.59	1.15	3.83
Mean± SD (mg/g)		42.65 ± 0.42	27.0 ± 0.3	50.6 ± 0.5	36.4 ± 0.5	33.0 ± 0.4	14.1 ± 0.1	0.28 ± 0.01	1.41 ± 0.06	0.86 ± 0.03
RSD_r_ (%) Interday		0.99	1.04	0.92	1.49	1.29	0.85	3.92	4.27	3.37
HorRat		0.62	0.60	0.59	0.91	0.78	0.45	1.15	1.60	1.17
		Root								
LOQ (mg/g)		0.151	0.160	0.092	0.229	0.063	0.141	0.090	0.021	0.101
D1	Mean± SD (mg/g)	20.2 ± 0.1	11.05 ± 0.05	25.9 ± 0.4	14.7 ± 0.1	18.8 ± 0.1	7.31 ± 0.05	0.128 ± 0.001	0.64 ± 0.01	0.52 ± 0.01
	RSD_r_ (%) Intraday	0.58	0.49	1.61	0.95	0.63	0.75	1.11	0.98	1.10
D2	Mean± SD (mg/g)	20.3 ± 0.1	11.11 ± 0.04	25.3 ± 0.2	14.8 ± 0.2	18.6 ± 0.2	7.33 ± 0.07	0.139 ± 0.003	0.70 ± 0.01	0.57 ± 0.01
	RSD_r_ (%) Intraday	0.31	0.38	0.87	1.14	1.01	0.97	2.14	1.51	1.39
D3	Mean± SD (mg/g)	20.00 ± 0.12	10.89 ± 0.06	24.68 ± 0.18	14.65 ± 0.08	18.55 ± 0.12	7.23 ± 0.06	0.137 ± 0.001	0.63 ± 0.01	0.514 ± 0.003
	RSD_r_ (%) Intraday	0.60	0.54	0.73	0.57	0.63	0.88	0.95	0.81	0.66
Mean± SD (mg/g)		20.19 ± 0.17	11.02 ± 0.11	25.30 ± 0.59	14.74 ± 0.15	18.62 ± 0.16	7.29 ± 0.07	0.135 ± 0.006	0.66 ± 0.03	0.53 ± 0.03
RSD_r_ (%) Interday		0.86	0.96	2.32	1.00	0.88	1.02	4.13	4.64	5.20
HorRat		0.48	0.49	1.33	0.53	0.48	0.49	1.09	1.55	1.69

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
