# Peer review of "A UHPLC-UV Method Development and Validation for Determining Kavalactones and Flavokavains in Piper methysticum (Kava)"

_molecules, 2019, doi:10.3390/molecules24071245_

Round 1

Reviewer 1 Report

Six kavalactones and three flavokavains were separated by UHPLC. The method was validated and used for quantification of the nine compounds in Kava roots, CO2 extracts, capsules and tablets.
I recommend it can be published after several minor comments:
Kava roots were extracted by acetonitrile, this can prevent or minimize the isomerization of yangonin and FKs, as authors mentioned.
However, based on Figure 4:
1) THY and DHY were found in all of roots, CO2 extracts, tablets and capsules. Then, does THY and DHY naturally exist in Kava roots? Or THY and DHY are artificial isomerization of yangonin?
2) Cis-FKA and Cis-FKB were found in tablets and capsules. Are they artificial isomerization products?
3) Was Cis-FKC found in tablets and capsules?
Above are essential for quality control. Anyway, these are just minor comments.

Author Response

The authors would like to thank the area editor and the reviewers for their precious time and invaluable comments. We have carefully addressed all the comments. The corresponding changes and refinements made in the revised paper are summarized in our response below.

(Note: The red word from Reviewer 1, and the blue one as our response)

1) THY and DHY were found in all of roots, CO2 extracts, tablets and capsules. Then, does THY and DHY naturally exist in Kava roots? Or THY and DHY are artificial isomerization of yangonin?

As the previous studies described (Ref. 26), both THY and DHY naturally exist in Kava raw material like Roots or any kava natural products. In our study, we believe that THY and DHY are naturally existed in Kava natural products.

Ref 26, Schmidt, A. H.; Molnar, I., Computer-assisted optimization in the development of a high-performance liquid chromatographic method for the analysis of kava pyrones in Piper methysticum preparations (vol 948, pg 51, 2002). Journal of Chromatography A 2006, 1110 (1-2), 272-272.

2) Cis-FKA and Cis-FKB were found in tablets and capsules. Are they artificial isomerization products?

In our study, the levels of FKA and FKB were relatively higher (ratio of FKs/KL) in the commercial tablets and capsules (bought from the market) than those in kava root and CO2 Extract (both from AFS). Since more processing procedures were needed for those commercial products, it is very likely that Cis-FKA and Cis-FKB could be formed by the isomerization of FKA and FKB during the whole processing from raw material (possible a higher FKA and B) to the final commercial products (tablets and capsules). Cis-FKA and Cis-FKB found in the commercial tablets and capsules were very likely as artificial isomerization products. The possible reasons could be a relatively higher level of FKA and FKB presented in the raw materials and some processing procedures included alcoholic solvents (or water) with light on.

3) Was Cis-FKC found in tablets and capsules?

Cis-FKC was not found or below detected level in the tablets and capsules, and also, the level of FKC was at a much lower level than FKA and FKB in those products.

Above are essential for quality control. Anyway, these are just minor comments.

Overall, as the conclusion for this study, 1st, this method maximized efficiency and chromatographic resolution under short analysis time. The natural minor KLs, DHY and THY, were well-separated from the major KLs under the excellent LC resolution. Then, the major KLs analysis is free of interference from minor KLs like THY & DHY. 2nd, The isomerizations of yangonin and FKs were prevented or limited by the usage of non-alcoholic solvent like acetonitrile for sample preparation in this method. This suggested method of kava analysis should be with a very low level interferences from cis-isomers of Yangonin and FKs. We believe that this method is a successful approach to determine major KLs and FKA, FKB, and FKC in kava raw materials, finished products (dry-filled capsule or formulated tablet), or a broad variety of market samples.

Reviewer 2 Report

Figures: peak abbreviations with their corresponding name of the separated compounds, conditions for HPLC analysis, detection (nm), UPHLC temperature, etc., should be indicated in figures captions.

Specify if the water was buffered and the water pH.

Results should be compared to other reports using similar conditions.

Author Response

The authors would like to thank the area editor and the reviewers for their precious time and invaluable comments. We have carefully addressed all the comments. The corresponding changes and refinements made in the revised paper are summarized in our response below.

(Note: The red word from Reviewer 2, and the blue one as our response)

Figures: peak abbreviations with their corresponding name of the separated compounds, conditions for HPLC analysis, detection (nm), UPHLC temperature, etc., should be indicated in figures captions.

Revised in the manuscript.

Specify if the water was buffered and the water pH.

Addressed in the section 2.1. The water is 100% HPLC or LC MS grade water, and pH is not applicable. 

Results should be compared to other reports using similar conditions.

Added in the discussion part

Round 2

Reviewer 2 Report

I have reviewed the revised version of the manuscript and it has been significantly improved and now warrants publication in Molecules.